# New Polyporphyrin Arrays with Controlled Fluorescence Obtained by Diaxial Sn(IV)-Porphyrin Phenolates Chelation with Cu^2+^ Cation

**DOI:** 10.3390/polym13050829

**Published:** 2021-03-08

**Authors:** Galina M. Mamardashvili, Dmitriy A. Lazovskiy, Ilya A. Khodov, Artem E. Efimov, Nugzar Z. Mamardashvili

**Affiliations:** G.A. Krestov Institute of Solution Chemistry of Russian Academy of Sciences, Akademicheskaya st. 1, 153045 Ivanovo, Russia; gmm@isc-ras.ru (G.M.M.); lazolvo@mail.ru (D.A.L.); ilya.khodov@gmail.com (I.A.K.); artem.efimov.1995@list.ru (A.E.E.)

**Keywords:** polyporphyrin arrays, chelation, fluorescence, hybrid materials

## Abstract

New coordination oligomers and polymers of Sn(IV)-tetra(4-sulfonatophenyl)porphyrin have been constructed by the chelation reaction of its diaxialphenolates with Cu^2+^. The structure and properties of the synthesized polyporphyrin arrays were investigated by ^1^H Nuclear Magnetic Resonance (^1^H NMR), Infra Red (IR), Ultra Violet - Visible (UV-Vis) and fluorescence spectroscopy, mass spectrometry, Powder X-Rays Diffraction (PXRD), Electron Paramagnetic Resonance (EPR), thermal gravimetric, elemental analysis, and quantum chemical calculations. The results show that the diaxial coordination of bidentate organic ligands (L-tyrazine and diaminohydroquinone) leads to the quenching of the tetrapyrrole chromophore fluorescence, while the chelation of the porphyrinate diaxial complexes with Cu^2+^ is accompanied by an increase in the fluorescence in the organo-inorganic hybrid polymers formed. The obtained results are of particular interest to those involved in creating new ‘chemo-responsive’ (i.e., selectively interacting with other chemical species as receptors, sensors, or photocatalysts) materials, the optoelectronic properties of which can be controlled by varying the number and connection type of monomeric fragments in the polyporphyrin arrays.

## 1. Introduction

Metal-coordination polymers are hybrid materials consisting of metal ions or clusters interconnected by rigid organic molecules (tectons) [1,2]. The ordering of the components in three dimensions, the possibility to use tectons of different natures and sizes, and the dynamic properties of the frameworks provide coordination polymers with unique luminescent, nonlinear optical, redox, magnetic, sorption, catalytic, ion exchange, sensory, and other properties [3,4,5,6,7,8,9,10] Due to their structure and unique physicochemical and photophysical characteristics, particularlytheir photoactivity, optoelectronic, and electrochemical properties, tetrapyrrole molecules are extremely promising objects for the construction of metal-coordination polymers for various purposes [11,12,13,14,15,16,17,18,19]. It is known that metal complexes of porphyrins and porphyrinoids are capable of selective reversible binding of substrate molecules and, thus, can be used to construct simple and complex supramolecular systems of various dimensions and architecture [20,21,22,23,24,25,26,27].

The aim of this work was to obtain new hybrid coordination oligomers and 1D- polymers with chelating binding of Sn(IV)-porphyrindiaxial complexes (*bis*-thyrazine-Sn(IV)-5,10,15,20-(4-sulfonatophenyl)porphyrin (I) and *bis*-diaminohydroquinone-Sn(IV)-5,10,15,20-(4-sulfonatophenyl)porphyrin (II)) with Cu^2+^ cations. Structures of the complexes I and II are depicted in Figure 1.

Complexes I and II were used as tectons coordinating through Cu^2+^ cations. Such ligands have the ability to form stable chelate cycles with *d*-metal cations due to the simultaneous interaction of the metal cation with the ligand reaction centers of different natures(the hydroxyl group oxygen and amino group nitrogen) [28]. Chain oligomerization of the Sn(IV)-porphyrindiaxial complexes (SnP(L)_2_) via Cu^2+^ cations is ensured by one copper cation forming two stable five-membered chelate rings, with the axial ligands belonging to the neighboring porphyrinates. The result of this oligomerization is the formation of stable nanoparticles (in comparison with less stable oligomers, which can be formed by four- or six-membered chelate rings based on copper cations), the sizes and properties of which depend on the nature of the axial ligands and the concentration ratio of the Sn(IV)-porphyrin axial complexes and *d*-metal cations. Coordination oligomers or polymers of this type are of particular interest to those involved in creating new ‘chemo-responsive’ (i.e., selectively interacting with other chemical species as receptors, sensors, or photocatalysts) or ‘size-responsive’ (i.e., capable of separating, storing, and transporting aggressive, toxic or explosive chemical species of different nature) materials, with their functional properties controlled by the number of monomeric fragments in the polyporphyrin arrays.

## 2. Materials and Methods

### 2.1. Materials

The high purity reagents were purchased commercially from PorphyChem (5,10,15,20-tetra(4-sulfonatophenyl)porphyrin tetraammonium), and Sigma Aldrich (St. Petersburg, Russia) (2,5-diaminohy droquinone dihydrochloride, L-tyrosine).

### 2.2. Equipment

All the ^1^H NMR (500.17) experiments were performed on a Bruker Avance III 500 NMR spectrometer (Bruker Biospin, Karlsruhe, Baden-Württemberg, Germany) with 256 or 512 scans and spectral windows of 20 ppm. The inaccuracy of the ^1^H NMR chemical shift measurement relative to the solvents (D_2_O and DMSO) was found to be ±0.01 ppm. The UV-Vis spectra were recorded in the range of 190–1200 nm on a JASCO V-770 spectrophotometer (Tokyo, Japan). The fluorescence spectra were recorded in the range of 430–770 nm on a Shimadzu RF 5301PC Spectrofluorimeter (Kyoto, Japan). The quantum-chemical calculations were performed using v.4.2.1 of the ORCA program system [29]. The Density-functional Theory (DFT) method with the CAM-B3LYP hybrid functional and 3–21 basis set was used to optimize the compound ground state. The pH was monitored by an Electroanalytical Analyzer (Type OP-300, Radelkis) ion meter. Elemental analyses were performed on a Flash EA 1112 analyzer. The mass spectra were obtained on a Shimadzu Biotech Axima Confidence Maldi TOF mass spectrometer of Kratos Analytical Limited-Great Britain, Manchester (with methanol as the solvent). The infrared analysis of the solid porphyrins was done on a VERTEX 80 V infrared Fourier-spectrophotometer (Ettlingen, Germany) with KBr pellets in the range of 4000–400 cm^−1^. The thermogravimetric analysis (TG) and differential thermal analysis (DTA) were recorded on a TG 209 F1 Iris thermomicrobalance (Netzsch, Germany) with dry samples at the heating rate of 10 C min^−1^ in an argon atmosphere in the range from room temperature to 900 °C. The Electron Paramagnetic Resonance (EPR) spectra of solutions in water were recorded on an EPR 10-MINI spectrometer (St. Petersburg) with an operating frequency of 9.45 GHz. The magnetic field was calibrated using a standard DPPH (diphenylpicrylhydrazyl) sample.

### 2.3. Synthesis

*Bis*-thyrazine-Sn(IV)-5,10,15,20-tetra(4-sulfonatophenyl)porphyrin (I) was obtained according to the procedure described by us previously in [30] from the bis-hydroxy-Sn(IV)-5,10,15,20-tetra(4-sulfonatophenyl)porphyrin (III). Mass-spectrum (MALDI-TOF): (m/z):[M+H]^+^ 1407.17; molecular formulaC_62_H_46_N_6_O_18_S_4_Sn-requires [M]^+^1406.01;UV-Vis (H_2_O), λ_nm_ (lgε): 594 (4.06), 555 (3.57), 421 (5.04), ^1^HNMR (500 MHz, D_2_O), ppm: 9.41 (s, 8H, Hβ-pyr.), 8.72 (s, 4H, NH_2_-L), 8.36 (d, J = 7.8, 8H, ortho-C6H4), 8.14 (d, J = 7.8, 8H, meta-C6H4), 5.51 (d, 4H, ortho-Ar-L), 4.37 (t, 2H, CH-L),3.19(m, 4H, CH_2_-L), 2.28 (d, 4H, meta-Ar-L);IR-spectrum, (KBr), ν, cm^−1^:3420 (sb)ν (OH), 3143 (b) ν (NH^3+^str.), 2939(w),ν(C-H, Ar), 2814(w),ν(C-H, Ar), 1680 (b)ν(C=C, Ar), 1655 (b) ν (NH^3+^deg. def.), 1607 (s)ν (COO- assym.), 1561(b)ν(C=C, Ph), 1517 (m)νNH^3+^sym. def.),1384 (s) ν (COO- sym.), 1367 bν(C=N), 1337 (w)ν (C-N, Por), 1246–45 (m) (NH^3+^rocking,),ν(C-N), 1200(w), (C-N, Pr), 1197 (m), 1128 (m), 1116(m)δ(C-H), 1045(m)ν(S-C), 1015(m) δ(C-H), 998 (m) ν(C-C),842–41, 744 (w)γ(C-H, Pyr), 706(w)γ(C-H, Ph), 706(w)γ(C-H, Ph), 646 (m) (COO- wagging), 580m(COO- rocking), 562 (m) ν(Sn-O).

*Bis*-diaminohydroquinone-Sn(IV)−5,10,15,20-tetra(4-sulfonatophenyl)porphyrin (II) was synthesized similarly to (I): 7.38 mg of III (0.0068 mmol) and 3.62 mg of 2,5-diaminohydroquinone dihydrochloride (0.017 mmol) were dissolved in 20 mL of distilled water. The resulting solution was boiled for 5 h, cooled, and then evaporated to dryness in a vacuum. The product was purified by column chromatography on neutral alumina using an ethanol-water mixture (1:2) as the eluent. The product yield after recrystallization was equal to 93%.Mass-spectrum (MALDI-TOF): (m/z):[M+H]^+^ 1325.39; molecular formula C_56_H_38_N_8_O_16_S_4_Sn-requires [M]^+^ 1324.01; UV-Vis (H_2_O), λ_max_ (logε) nm:419 (5.11),554 (4.10), 593 (3.61);^1^H NMR, (500 MHz, D_2_O): 9.10 (s, 8H,β-pyrr.), 8.45 (d, J = 7.8 Hz, 4H, ortho-C_6_H_5_), 8.25 (d, J = 7.7 Hz, 8H, meta-C6H5), 8.59 (s, br, 2H, NH_2_ (L)), 5.32 (s, br, 2H, NH_2_ (L)), 5.97(t, J = 8.0 Hz, 2H, Ar(L)), 4,90 (s, 2H, OH(L)), 2.92 (t, J = 2.0 Hz, 2H (L)).IR-spectrum, (KBr), ν, cm^−1^:3357 (sb)ν (N-H), 3244 (sb)ν (O-H) ν, 3052,2930-ν (C-H, Ar), 1695(b)ν(C=C, Ar), 1619(N-H)δ, 1582(b)ν(C=C, Ph), 1601, 1501, 1478 (C-C, Ar) ν, 1381 bν(C=N, Por), 1359 (w)ν (C-N, Por), 1152 (C-O)ν, 1045(m)ν(S-C), 1015(m) δ(C-H), 998 (m) ν(C-C), 821, 7 50 (C-H) γ, 784 (N-H) γ_w,_ 699 (C-C)γ, 566 (m) ν(Sn-O).

*Bis-*hydroxy-Sn(IV)-5,10,15,20-tetra(4-sulfonatophenyl)porphyrin (III) was synthesized according to the method described by the authors of [31]. Mass-spectrum (MALDI-TOF): (m/z):[M+H]^+^ 1081.23; molecular formula C_44_H_26_N_4_O_14_S_4_Sn-requires [M]^+^ 1080.02;UV-Vis (H_2_O), λ _max_(logε) nm: 593 (4.10), 554 (3.60), 419 (5.40), ^1^H NMR, (500 MHz, D_2_O): 9.10 (s, 8H,β-pyrr.), 8.45 (d, J = 7.8 Hz, 4H, ortho-C_6_H_5_), 8.25 (d, J = 7.7 Hz, 8H, meta-C_6_H_5_). −7.02 (2H, OH).

The synthesis of dimeric (I-Cu-I, II-Cu-II), oligomeric (Cu-[I-Cu]_6_ and Cu-[II-Cu]_6_) and polymers ([I-Cu]_n_ and [II-Cu]_n_) porphyrins was carried out by heating an aqueous solution of the corresponding axial complex I or II and copper chloride. The concentration of the complexes was at least 5 × 10^−4^ mol/L.

Synthesis of I-Cu-I, Cu-[I-Cu]_6_ and [I-Cu]_n_: 13.5 mg (0.0096 mmol) of complex I was dissolved in 10 mL of distilled water. Then, 1.63 mg (0.0096 mmol) or 8.2 mg (0.0480 mmol) of copper chloride dihydrate was added to the resulting solution to obtain a molar ratio of I-Cu^2+^ equal to 1:1 or 1:5, respectively. To suppress hydrolysis, the reaction mixture was acidified with several drops of diluted hydrochloric acid. The resulting reaction mixtures were heated for 24 h at a temperature of 85–90 °C. After the reaction was completed, the soluble and insoluble reaction products were separated by filtration at atmospheric pressure. The insoluble reaction product ([I-Cu]_n_) was repeatedly washed with distilled water on a filter.

Synthesis of II-Cu-II, Cu-[II-Cu]_6_,and [II-Cu]_n_: 12.5 mg (0.0094 mmol) of complex II was dissolved in 10 mL of distilled water. Then, 1.6 mg (0.0094 mmol) or 8.0 mg (0.0471 mmol) of copper chloride dihydrate was added to the resulting solution to obtain a molar ratio of II-Cu^2+^ equal to 1:1 or 1:5, respectively. The rest of the procedure was similar to the synthesis of polymers and oligomers of I with Cu^2+^.

## 3. Results and Discussion

### 3.1. Synthesis and Structure

It is well known that when amino acids, such as some other polydentate ligands, interact with *d*-metal cations, they form stable compounds with one or two chelate rings [28]. The higher stability of such compounds is the result of each polydentate ligand binding to the complexing cation by at least two bonds (-M-O, M ← NH_2_ or M ← NH). The products of the amino acid interaction with *d*-metal cations can be mono- and bis-ligand particles. In the latter case, bicyclic chelating of the copper cations occurs with formation of 4-coordinate square planar geometry of the coordination center [32,33].

Depending on the self-assembly conditions, the products of the interaction of *bis*-axial complexes I and II with the Cu^2+^ cations can be both porphyrin dimers ([I-Cu-I] and [II-Cu-II]) and oligomers ([I_n_-Cu_n±1_] and [II_n_-Cu_n±1_]) consisting of several porphyrin fragments and copper cations (Figure 2).
(1)$$\mathrm{NSnP}(L{)}_{2}+(n\pm 1){\mathrm{Cu}}^{2+}\overset{t}{\to }[\mathrm{SnP}(L{)}_{2}{]}_{n}-{\left[\mathrm{Cu}\right]}_{n\pm 1}$$

The structures of *bis*-axial complexes I and II and products of their self-assembly (porphyrin dimers (I-Cu-I and II-Cu-II)), obtained by simultaneous interaction of Cu^2+^ with the hydroxy and amino groups of axial ligands belonging to two different porphyrinate molecules, were optimized by the DFT method with the CAM-B3LYP hybrid functional and 3–21 basis set. The data obtained are shown in Figure 3 and Table 1.

As seen from Figure 3 and Table 1, complexes I and II had similar Sn-O and Sn-N bond lengths. A distinctive feature of I wasthe presence of additional points of binding between the axial ligands and the porphyrin macrocycle due to the formation of intramolecular hydrogen bonds, which could potentially prevent the formation of oligomeric and polymer structures. The inclination angle of the axial ligand aromatic part of the axial ligand to the porphyrin plane in complex I was 41°, whereas in complex II, it was 50°.

The formation of dimeric structures increased the inclination angle of the ligand phenolate fragment relative to the porphyrin plane, probably due to the repulsion of the aromatic fragments from each other. The functional groups involved in the chelation with Cu^2+^ werelocated in the dimeric structures at the maximum possible distance from the porphyrin plane. Obviously, in the case of a two-center interaction of the axial fragments with Cu^2+^, such a structure is the most favorable energetically. In the case of I-Cu-I, the formation of a chelate bond between the tyrosine and the copper cation destroys the hydrogen bonds between the tyrosine and sulfophenyl moieties.

A significant increase in the Sn-O-L angle can be observed in the II-Cu-II structure optimized by quantum chemical calculations. This increase is associated with the fact that the amino group of the diaminohydroquinone fragment approached the pyrrole nitrogen atom of the porphyrin macrocycle. Since there can be a significant electrostatic interaction between the porphyrinate nitrogen atom and the amino group protons, such a structure distortion can be energetically favorable.

Since the axial ligands in complexes I and II were of different sizes, the distance between the porphyrin fragments in the I-Cu-I and II-Cu-II dimers differed significantly and amounted to 21.3 and 17.6 Å, respectively. At the same time, the porphyrin fragments in the dimeric systems were almost parallel to each other (Figure 3). The structures of the porphyrin oligomers linked through Cu^2+^ werenot optimized. However, based on the data about the dimeric structures, it can be assumed that the longer porphyrin oligomers were almost linear, and the porphyrin polymers consisted of fragments similar to those shown in Figure 3.

According to the experimental data, the result of these self-assembly of Sn(IV)-porphyrinates (I-II) in the presence of Cu^2+^ in aqueous solutions depends on the concentration ratio of the starting reagents, reaction time, and temperature. Table 2 shows the empirical formula, molecular weight, and elemental analysis data of the reaction (1) products at different concentrations of the starting compounds. Oligomerization was achieved by heating compounds I and II for several hours at 90 °C. The self-assembly of the porphyrinate fragments was monitored by changes in the UV-Vis spectra.

The self-assembly of the porphyrinate macrocycles into larger aggregates led to a decrease in their solubility. The larger the oligomer, the lower its solubility. Upon reaching a certain size, the resulting oligomers precipitated. The proportions of soluble and insoluble self-assembly products in the studied systems are also presented in Table 2.

An analysis of the molecular weights of the substances presented in Table 2 shows that the soluble products of the interaction of I or II with Cu^2+^ at an equivalent quantitative ratio of the reagents were mainly porphyrin dimers (I-Cu-I and II-Cu-II). Under the conditions of a five-fold excess of copper cations and prolonged heating of the reaction mixture (up to 24 h), oligomers with a large number of macrocycles were formed. The maximum number of porphyrin fragments in soluble oligomers didnot exceed six. Chain oligomers with more than six Sn(IV)-porphyrin units (polymers) precipitated during reaction (1). The formation of porphyrin oligomers and polymers through strong bis-chelate binding with the formation of a flat coordination center (Figure 2) was confirmed by UV-vis, IR, ^1^H NMR, EPR spectroscopy, and thermogravimetric analysis. The composition of oligomeric chains was estimated from the data of elemental analysis, mass spectrometry, and 2D NMR.

The mass spectrometry confirmed the formation of dimeric forms of complexes I and II in the products of reaction (1). In addition to the peaks with m/z 1406.01 and 1470.51, corresponding to the [I-H]^−^ and ([I-Cu]-H)^−^ ions, the mass spectrum of the product of the complex I interaction with Cu^2+^ (Figure 4) at a 1:1 molar ratio of the reagents hada peak with m/z 2877.03 corresponding to the [I-Cu-I] dimer. It was not possible to confirm the formation of larger (containing six macrocyclic fragments)porphyrin oligomers by the mass spectrometry method, which was probably due to the oligomer instability in the conditions of the mass spectral studies of the samples. Similar behavior wasobserved in the mass spectra of the products of the complex II interaction with Cu^2+^ at 1:1 and 1:5 molar ratios of the reagents.

### 3.2. Thermogravimetric Analysis and Powder XRD

All the products of reaction (1) were thermally stable solids, indicating a strong metal–ligand bonding. Figure 5 shows DTA and TG curves with endo- and exothermic peaks of complex I and the oligomer based on it. The thermal behavior of the free molecules of aminoacids, including tyrazine, has been well studied. According to the results found by the authors of [34,35], the first endothermic stage of tyrazine decomposition occurs in the temperature range of 276–322 °C and corresponds to the reactions of its decarboxylation and deamination. Further, in the temperature range of 322–350 °C, the resulting intermediate product is relatively stable. The second stage of tyrazine decomposition occurs at 350–355 °C and involves oxidation of the phenolic fragment. At the same time, the porphyrin macrocycles in the general case [36], particularly the Sn(IV)-porphyrins [37], are highly resistant to thermal oxidative destruction.

As Figure 4 shows, the decomposition of complex I consisted of four stages. At the first stage, in the temperature range up to 100 °C, the complex thermal dehydration occurred. The loss of 9.45% of the sample mass corresponded to water evaporation as the sample itself was not subjected to preliminary drying. At the second stage, in the temperature range of 217–441 °C, the loss of 15.8% of the sample mass indicates partial axial ligand decomposition. The third stage, in the temperature range of 518–727 °C, the loss of 29.2% of the sample mass corresponded to the detaching of four sulfo groups from the porphyrin aryl fragments. The fourth stage consisted of the removal of the phenyl fragments, both of the porphyrin and axial ligands (15.5% of the sample mass). The residue mass (30% of the sample mass) indicates that the tetrapyrrole macrocycle containing the Sn(IV) cation in the coordination center was not destroyed. Similar data on the thermal decomposition of Sn(IV)-porphyrins have been described by the authors of [37]. The first stage of decomposition of the tyrazine fragments in porphyrin complex I begins at lower temperatures than that of the free aminoacid ligand, whereas the second stage (phenyl fragment oxidation), on the contrary, occurs at a higher temperature.

A thermal analysis of the hexamers shows that decomposition of this compound consists of more stages. The first stage (up to 100 °C), as in the case of monomeric complex I, was associated with thermal dehydration. The second stage, in the temperature range of 100–136 °C, consisted of the dehydration of the water molecules located in the coordination bis-chelate center of Cu^2+^. The third stage of destruction, in the temperature range of 213–351 °C, corresponded to the destruction stage of the tyrazine fragments of the chelate cycles. At the next stage, in the temperature range of 434–605 °C, the porphyrin macrocycle sulfo groups were eliminated. The last stage, in the temperature range of 736−925 °C, was probably associated with the processes of removal of the phenyl fragments of the porphyrin macrocycle and axial ligands. The residue mass (33% of the initial sample mass) indicates that the residue contains Sn(IV)-porphyrin and CuO. Similar results were obtained for the [I-Cu]_n_ polymer. The data for the II and its hexamers are depicted in Table 3. 

Powder X-ray diffraction (PXRD) was performed to verify the purity of [I-Cu]_n_ and [II-Cu]_n_. The PXRD curve of [II-Cu]_n_, shown in Figure 6 as an example, indicates a diffuse large steam bun peak. The PXRD curve of [I-Cu]_n_ looks similar. The absence of other obvious sharp peaks in the corresponding curves indicates that the polymers wereamorphous, with random growth during the self-assembly [38].

### 3.3. UV-Vis and IR-Spectral Studies

The UV-Vis spectra of the water-soluble products of reaction (1) were recorded in the UV-visible region (Figure 7 and Table 4). The spectra for the investigated copper(II) complexes displayed bands at 610 nm and 661nm, assigned to ^2^B_1g_→^2^E_g_ and ^2^E_g_ → 2A_1g_*d-d* transitions. According to the authors of [39,40,41], this indicates that the investigated complexes weremononuclear complexes with four-coordinate square planar geometry.

The Fourier Transform Infrared (FTIR) spectra of the metal complexes were recorded in KBr discs over the range of 4000–400 cm^−1^. The data of the IR studies (Table 5 and Figure 8) of the corresponding samples provide valuable information on how axial complexes I and II bind to Cu^2+^ during the formation of chelate complexes. Based on the analysis of the spectra of the reaction (1) products, it can be concluded that the amino and carboxyl groups were simultaneously involved in the chelate complex formation. The IR spectra of the oligomers now have new bands caused by the bending vibrations of the bonds formed due to the coordination with Cu^2+^. The frequency ranges expected for these vibrations are well known [42]. In addition to the vibrations of the amino and carboxyl groups, the processes of chelation were also confirmed by the vibrations of the N-M and O-M bonds.

The IR spectra of the aminoacid fragments with a bipolar structure contained characteristic bands of the NH^3+^-group corresponding to symmetric stretching (in the region of 3200–3400 cm^−1^) and bending (in the region of 1550–1600 cm^−1^) vibrations. In the chelate complexes, the stretching vibrations of the bound NH_2_ group were shifted to longer wavelengths. Such a decrease in the frequency and increase in the intensity of the amino-group stretching vibrations can be interpreted by coordination interactions between the metal cation and the nitrogen atom of the amino-group, which increased the dipole moment value. Also characteristic of chelation is the band at 1160 cm^−1^, which was related to the deformation vibrations of the NH_2_ group but was not observed in the bipolar compound.

A vibration band typical of the free carboxylate anion appeared at 1607 cm^−1^ and 1384 cm^−1^. The carboxyl group transition to the non ionized state caused this band to disappear, and the vibration appeared in the longer wavelength region as ν(C=O) in the carboxyl group. For the investigated complexes, the COO−asymmetric stretching frequencies were shifted to lower values compared with those of the ligand. The bands in the region of 480 cm^−1^ indicate the formation of a Cu–O bond and further confirm the ligand coordination to the central metal ion via the oxygen atom of the carboxylate group [42]. Hypsochromic shifts were observed for the –NH_2_ frequencies during coordination. This indicates bond elongation during the coordination, therefore suggesting probable square planar geometry of the complexes. The new bands in the spectra of the complexes at 535–552 cm^−1^ were assigned to the (M–N) stretching frequency. The participation of the lone pairs of electrons on the N atom of the amino group in the ligand in the coordination was confirmed by these band frequencies [43].

### 3.4. EPR Studies

The conclusions about the planar-square structure of the obtained Cu(II) complexes based on the results of the IR spectra were additionally confirmed by EPR spectroscopy data [44,45,46,47]. In the EPR spectra of the studied compounds (Figure 9) at room temperature, the hyperfine lines from the magnetic interaction of the unpaired electron spin with the copper atom nuclear spin were well resolved. The isotropic EPR spectra are described by a symmetric spin Hamiltonian and had four hyperfine lines of equidistant components of different intensities and widths for nuclear spin projections, which is explained by the McConnell relaxation mechanism [47]. The spectra were a superposition of the spectra from the ^63^Cu nuclei, with the trans-N_2_O_2_ coordination environment of the Cu(II) ion.

The presence of two five-membered metallocycles in complex compounds, regardless of the nature of the coordinated atoms, led to a planar conformation. The coordination center in oligomers based on II increased the electron-donating properties of the nitrogen and oxygen atoms. These conclusions were confirmed by the calculated parameters of the EPR spectra. The EPR parameters for the Cu- [I-Cu]_6_ hexamer with tyrazine fragments (L1), had the following values: g = 2.119, a_cu_ = 89.6 E, α^2^ = 0.81, whereas for the oligomer with aminoresorcinol ligands, these values were within the following range: g = 2.108, a_Cu_ = 95.84 E, α^2^ = 0.89. The αparameter calculated from the isotropic EPR parameters using Formula (2) [48]:(2)$$\alpha^{2}=\frac{1}{0.43}\left(\frac{\alpha_{Cu}}{0.036}+g-2\right)+0.02$$
characterizes the degree of covalence of the copper-ligand bond. If the oligomer based on II had α^2^ = 0.89, then the oligomer based on I was somewhat lower (0.81).

### 3.5. NMR Spectroscopy Studies

The NMR spectroscopy is a very important tool for the investigation of the structure of an unknown compound in solutions. Data of two-dimensional ^1^H NMR make it possible not only to obtain information confirming the presence of chelate binding in the products of reaction (1), but also to determine the number of porphyrinate fragments in the resulting porphyrin oligomers. The formation of chelating bonds of porphyrinate axial ligands with Cu^2+^ is evidenced by characteristic shifts in the signals of the ligand protons located in close proximity to the inner coordination sphere of the copper cations. The NMR study results are presented in Table 6.The absence of signals of protons of the -COOH and -OH groups indicates the formation of the corresponding Cu(II)-complexes (due to the replacement of H^+^ with the metal ion). The signal of the protons at the carbon atom, which was closer to the NH_2_ group, was significantly shifted (by 0.5 ppm) in a strong field.

Diffusion-ordered spectroscopy (DOSY) was used to determine the composition of the reaction (1) products between Sn(IV)-porphyrin axial complexes and Cu^2+^. It has been reported in recent works [30,49,50,51,52,53,54] that this method is among the most effective in the analysis of supramolecular complexes of macrocyclic compounds. This method makes it possible to confirm the structures of the formed supramolecular complexes by comparing the diffusion coefficients of the systems obtained by self-assembly with the diffusion coefficients of the initial compounds (before the self-assembly) taken as objects of comparison. In our case, diaxial complexes I and II were employed as the reference compounds. The diffusion coefficients (D) of complexes I and II and the products of their interaction with Cu^2+^ (in 1:1 and 1:5 ratios) were measured by the stimulated echo method, with a bipolar gradient and a WATERGATE pulsed water suppression unit [55] in an H_2_O/D_2_O mixture (in a 90:10 ratio) at 298 K. The results are presented in Table 7 and Figure 10.

The high accuracy of these measurements clearly indicates that the DOSY method is sensitive enough for us to speak with confidence about the difference between the complexes of the monomeric porphyrinates and oligomeric porphyrin systems and to confirm the complexation process in the studied systems.

For the sake of simplicity of interpretation of the DOSY experiments, we conducted a graphical analysis, which has been successfully applied to related/similar molecular systems earlier [56,57,58]. This graphical analysis is based on a model of a mass dependence on the coefficient of translational diffusion, obtained from the Einstein–Smoluchowski relation [59,60]. Thus, it is shown that the ratio of the diffusion coefficients for two different molecular particles (D_i_/D_j_) is inversely proportional to the square root or cubic root of the ratio of their molecular masses (M_j_/M_i_) for rod-like and spherical forms of molecules, and can be calculated by the formula:(3)MjMi2≥ DiDj≥ MjMi3

This ratio can be used to calculate a set of theoretical diffusion coefficients (upper and lower limits) for each supramolecular complex based on the diffusion coefficients of starting complexes I and II (monomers). As shown by Cabrita and Berger [61], the use of a reference compound is effective for solving problems associated with qualitative and quantitative analysis of intermolecular interactions. For graphical analysis, in addition to the theoretical curves of the solvent diffusion coefficients shown in Figure 11 (the black line refers to the simulated theoretical dependence for rod-shaped oligomeric particles, the dotted line refers to the simulated theoretical dependence for spherical oligomeric particles), we indicated the experimental values of the self-diffusion coefficients determined both for initial complexes I and II and the products of their interaction with Cu^2+^. The performed graphical analysis showed that the experimental values of the diffusion coefficients of the reaction (1) products at 1:1 and 1:5 ratios of the starting compounds fit well in the range of the calculated theoretical curves. The data obtained indicate that the products of reaction (1), with an equivalent ratio of reactants in the case of both complex I and complex II, were most likely dimers with molecular weights of 2877.57 g/mol (I-Cu-I) and 2715.37 g/mol (II-Cu-II). The systems formed with a five-fold excess of copper cations were characterized by the formation of Cu-[I-Cu-]_6_ oligomers with molecular weights of 8892.89 g/mol and Cu-[II-Cu-]_6_ oligomers with molecular weights of 8400.3 g/mol.

### 3.6. Fluorescent Properties Studies

Figure 12 shows the change in the fluorescent properties of complexes I and II as the corresponding dimers and oligomers were formed from them. The distinguishing feature of the presented spectra was an additional peak in the region of 620–625 nm as the corresponding porphyrin arrays with different numbers of tetrapyrrole chromophores (n = 2, 6) were formed from the porphyrin monomers (complexes I and II). Such a peak probably appeared because the porphyrin dimers and oligomers formed during the chelation had an additional energy level, enabling an emitting transition to the ground state.

Strong quenching of the fluorescence (Figure 11) of complexes I and II in comparison with bis-hydroxy-5,10,15,20-tetra-(4-sulfonatophenyl)porphyrin-Sn(IV) (III), according to the literature data [62,63] and the results of our own studies [64,65], is caused by the interaction of the closely spaced aromatic systems of the ligand and porphyrin macrocycle (in complex I, the inclination angle of the axial ligand aromatic part to the porphyrin plane was31°, while in complex II, it was 50°). The results of the quantum chemical calculations show (Figure 2, Table 1) that as dimeric structures were formed, the inclination angle of the phenolate fragment of the ligands relative to the porphyrin plane increased (the angle became close to 90°). The functional groups involved in the chelation with Cu^2+^ cations in the dimeric structures were located at the maximum possible distance from the porphyrin plane. It is logical to assume that the structural changes accompanying the formation of dimeric and oligomeric systems weakened the mutual influence of the aromatic systems of the ligand and macrocycles in them. This is in good agreement with the data presented in Figure 13. In the case of complex I, the quantum yield of the systems formed at different ratios of the reagents (1:1 or 1:5) increased by about two-fold. In the case of complex II, at a 1:5 molar ratio of the reagents, the quantum yield of fluorescence increased by about four-fold. The difference in the quantum yields of the dimeric and oligomeric systems obtained on the basis of complexes I and II could probably be explained by the different sizes of the axial ligands in the corresponding complexes. The importance of spatial effects was confirmed by the data in Tabl. 1, according to which the distance between the porphyrin fragments in the I-Cu-I and II-Cu-II dimers differed significantly and amounted to 21.3 and 17.6 Å, respectively.

It should be also noted that some of the products of reaction (1) precipitated. It is logical to assume that the polymer products of the reaction of the Cu^2+^ cation chelate complex formation with the studied axial complexes of Sn(IV)-porphyrin were precipitated. Currently, our laboratory is conducting research related to the establishment of their structure and properties. According to the preliminary studies, these porphyrin polymers are characterized by high porosity and capacity to selectively adsorb organic solvent molecules. This suggests that coordination polymers of this type could be promising “size-responsive” materials (i.e., capable of separating, storing, and transporting aggressive, toxic, or explosive chemical species of different natures).

## 4. Conclusions

Thus, the obtained porphyrin oligomers and polymers in solid state and in solution are compounds in which the porphyrin fragments with tyrosine and diaminoresorcinol axial ligands form stable coordination compounds with two five-membered square planar metallocycles. Soluble products of the chelation of Sn(IV)-tetra(sulfonatophenyl)porphyrin diaxial complexes with Cu^2+^ are porphyrin coordination oligomers with different numbers of tetrapyrrole fragments (from two to six). The specific composition of the interaction products depends on the concentration ratio of the reagents. If, at an equivalent concentration ratio of the reagents, the main products are porphyrin dimers, then an excess of Cu^2+^ leads to the formation of larger oligomeric porphyrin arrays. The obtained porphyrin oligomers formed by five-membered chelate rings with Cu^2+^ are stable compounds (in comparison with oligomers, which can be formed by four- or six-membered chelate rings based on copper cations). The results show that chelation of Sn(IV)-porphyrin diaxial complexes with Cu^2+^ is accompanied by an increase in the fluorescence of the resulting hybrid organic-inorganic oligomers. The results obtained are of particular interest to those involved in creating of new ‘chemo-responsive’ (i.e., selectively interacting with other chemical species as receptors, sensors, or photocatalysts) materials, the optoelectronic properties of which could be controlled by varying the number of monomeric fragments in the polyporphyrin arrays. 

## Figures and Tables

**Figure 1 polymers-13-00829-f001:**
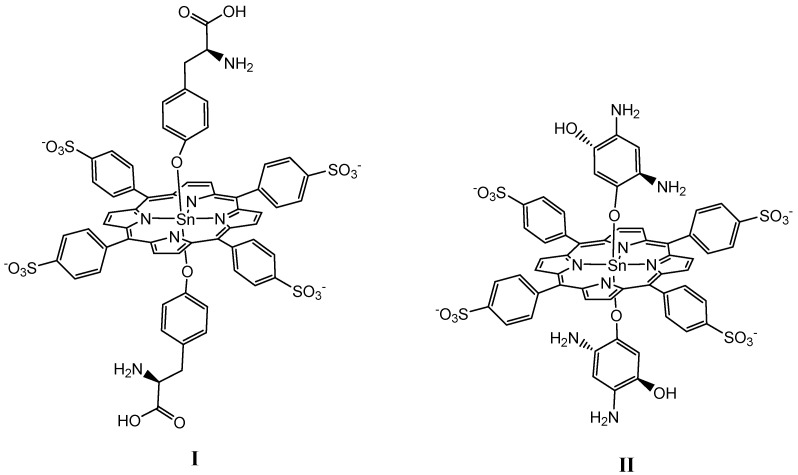
Structures of the complexes I and II.

**Figure 2 polymers-13-00829-f002:**
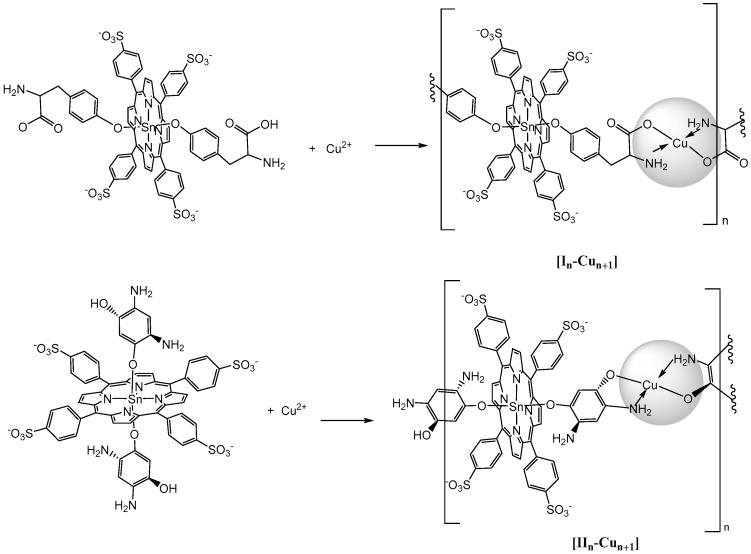
Proposed structures of products of the of the Sn(IV)-porphyrin axial complexes I and II interaction with Cu^2+^ cations.

**Figure 3 polymers-13-00829-f003:**
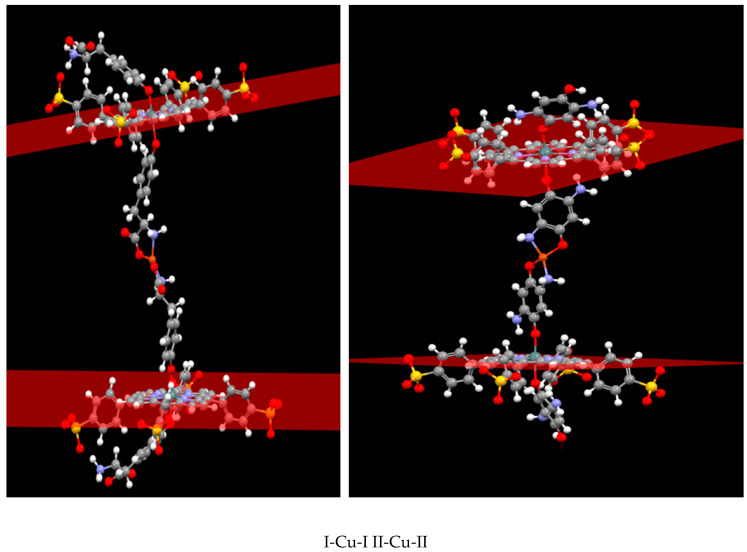
Structures of the dimers I-Cu-I and II-Cu-II optimized by the DFT/CAM-B3LYP hybrid functional and 3–21 g basis.

**Figure 4 polymers-13-00829-f004:**
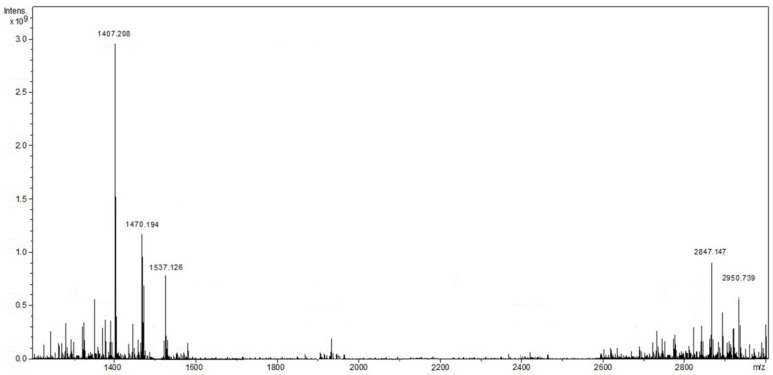
Mass spectrum of the I-Cu-I.

**Figure 5 polymers-13-00829-f005:**
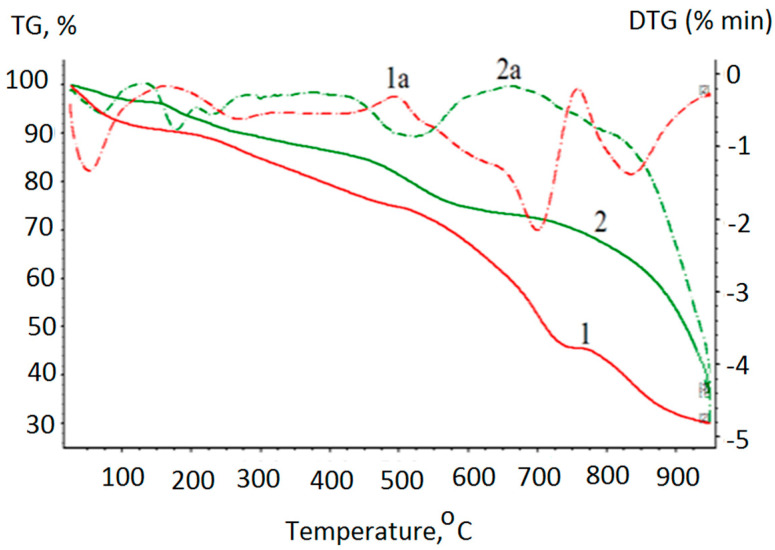
Differential thermal analysis (dashed line, DTA) and thermogravimetric analysis (solid line, TG) curves with endo- and exothermic peaks for thermal decomposition of I (red line) and Cu-[I-Cu]_6_ (green line).

**Figure 6 polymers-13-00829-f006:**
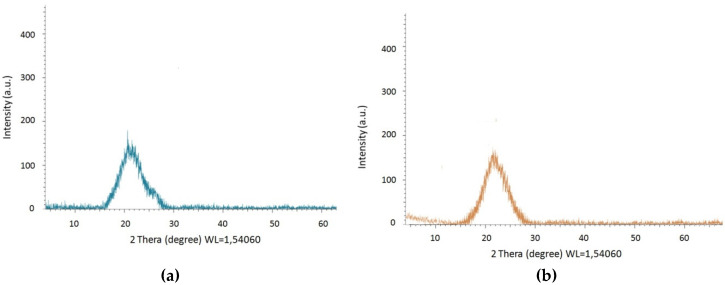
Powder X-ray diffraction XRD (PXRD) of the [I-Cu]_n_ (**a**) and [II-Cu]_n_ (**b**).

**Figure 7 polymers-13-00829-f007:**
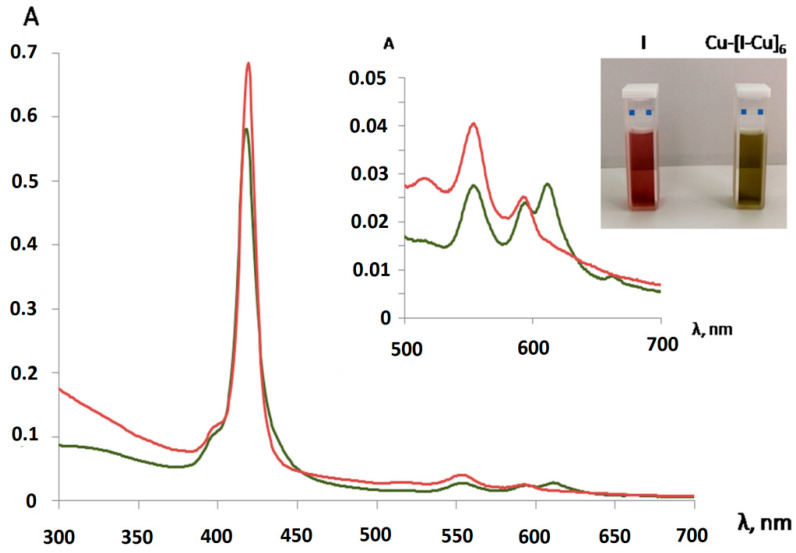
UV-Vis spectra of complex I (red line) and hexamers Cu-[I-Cu]_6_ (green line) in water.

**Figure 8 polymers-13-00829-f008:**
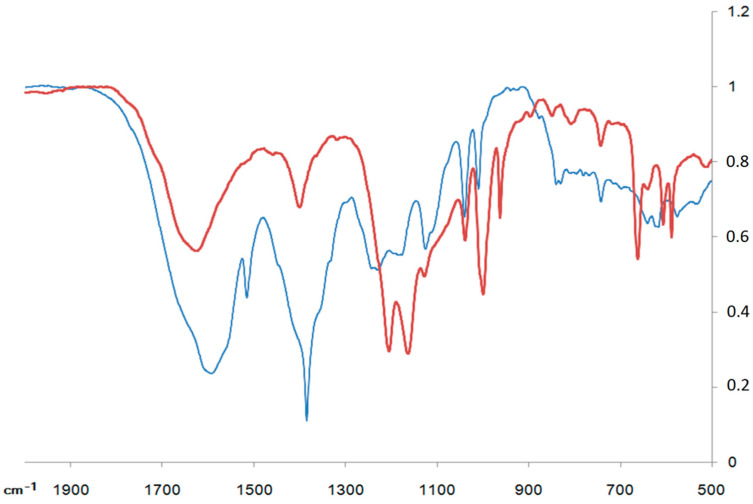
IR spectra of complex I (blue line) and Cu-[I-Cu]_6_ (red line) in KBr discs.

**Figure 9 polymers-13-00829-f009:**
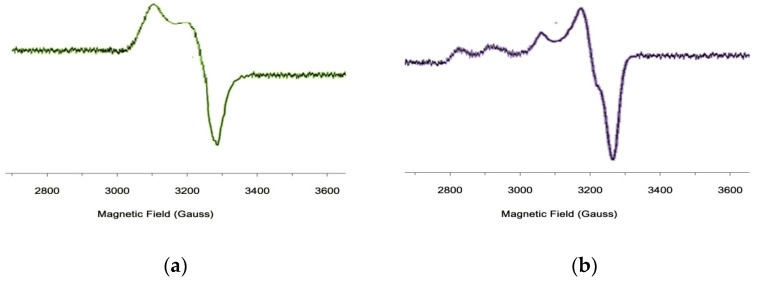
Powder Electron Paramagnetic Resonance (EPR) spectrum of Cu-[I-Cu]_6_ (**a**) and [I-Cu]_n_ (**b**).

**Figure 10 polymers-13-00829-f010:**
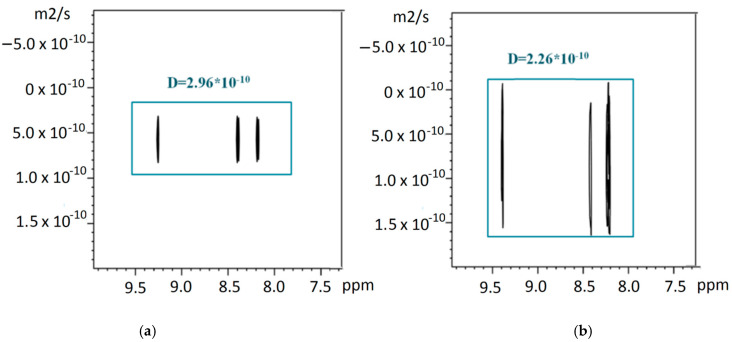
^1^H NMR diffusion-ordered spectroscopy (DOSY) spectra of products of interaction the complex I (**a**) and porphyrin dimers with Cu^2+^I-Cu-I (**b**).

**Figure 11 polymers-13-00829-f011:**
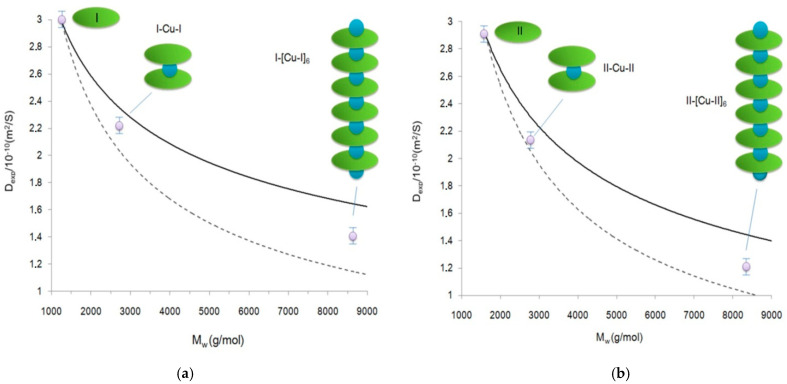
Graphical analysis of self-diffusion coefficients of the products of SnP(L)_2_ interaction with Cu^2+^ cations at the 1:1 and 1:5 ratios with monomer complexes taken as the reference standard: (**a**)-I, (**b**)-II. The solid lines represent the theoretical values calculated by the Formula (3).

**Figure 12 polymers-13-00829-f012:**
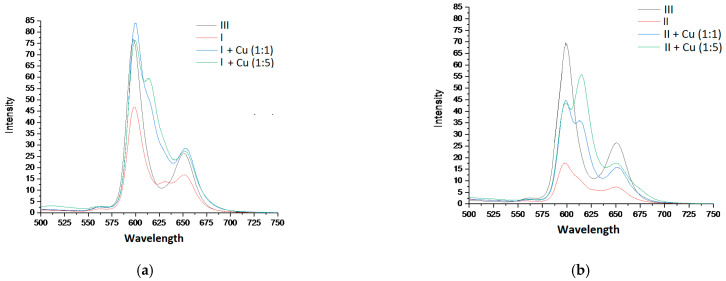
Fluorescence spectra of the studied systems with different concentration ratios of the reagents, λ_ex_ = 416 nm (I-(**a**), II-(**b**)).

**Figure 13 polymers-13-00829-f013:**
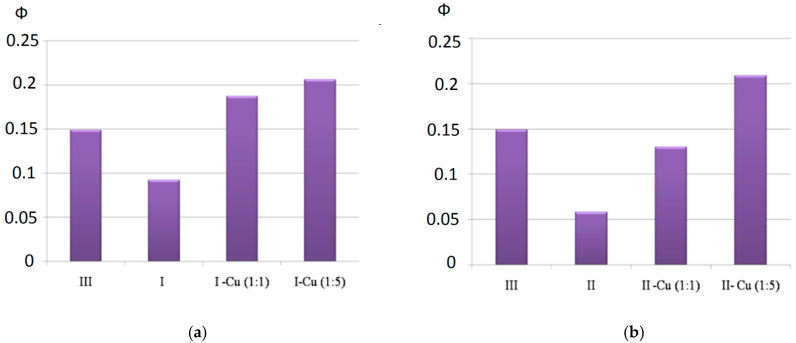
(**a**) Fluorescence quantum yields of complexes III and I and products of their interaction with Cu^2+^ depending on the concentration ratio of the reagents; (**b**) fluorescence quantum yields of complexes III and II and products of their interaction with Cu^2+^ depending on the concentration ratio of the reagents.

**Table 1 polymers-13-00829-t001:** Geometric parameters of the studied compounds obtained by quantum-chemical calculations using the Density-functional Theory DFT/CAM-B3LYP hybrid functional and 3–21 g basis.

Compounds	I	I-Cu-I	II	II-Cu-II
The maximum distance from the upper point of the ligand to the porphyrin core, Å	7.061	10.39314	6.560	7.3955
r(Sn-O), Å	2.0517	2.0517	2.0517	1.9902
r(Sn-N), Å	4.238	4.1662	4.2244	4.1778
r(Cu-O), Å	-	1.81772	-	1.8220
r(Cu-N), Å	-	1.92909	-	1.9377
<L-O-O-L(Ligand rotation angle)	98°	13° and 97°	159°	25° and 149°
<Sn-O-L (The bridge angle)	122°	145°	131°	172°
The angle between porphyrin end aromatic ligand planes	41°	41° 70°	50 °	50° 87°
The angle between the porphyrin planesin the dimer	-	9°	-	9°

**Table 2 polymers-13-00829-t002:** Empirical formula, molecular weight, and elemental analysis data of the reaction (1) products with the ratio of reagents (1:1 and 1:5).

Compounds	Yield,%	Formula	Found/Calcd			
			Cu	C	H	N
I	-	C_62_H_44_N_6_O_18_S_4_Sn 1407.01	-	52.89	3.15	5.97
II	-	C_56_H_38_N_8_O_16_S_4_Sn 1325.91	-	50.73	2.89	8.45
I: Cu (1:1)	94%	C_62_H_44_N_6_O_18_S_4_SnCu_0.5_ I-Cu-I 2877.57	2.19/ 2.21	51.67/ 51.72	3.06/ 3.08	5.81/ 5.84
II: Cu (1:1)	96%	C_56_H_38_N_8_O_16_S_4_SnCu_0.5_ II-Cu-II 2715.38	2.32/ 2.34	49.40/ 49.54	2.80/ 2.82	8.22/ 8.25
I: Cu (1:5) ^a^	78%	C_62_H_44_N_6_O_18_S_4_SnCu_1.17_ Cu-[I-Cu]_6_ 8892.89	4.98/ 5.00	50.48/ 50.24	2.96/ 2.99	5.64/ 5.67
II: Cu (1:5) ^a^	84%	C_56_H_38_N_8_O_16_S_4_SnCu_1.17_ Cu-[II-Cu]_6_ 8400.31	5.27/ 5.30	47.98/ 48.04	2.72/ 2.74	7.97/ 8.00
I: Cu (1:5) ^b^	22%	C_62_H_44_N_6_O_18_S_4_SnCu [I-Cu]_n_ n× [1471.56]	4.27/ 4.32	50.62/ 50.60	3.00/ 3.014	5.68/ 5.71
II: Cu (1:5) ^b^	16%	C_56_H_38_N_8_O_16_S_4_SnCu [II-Cu]_n_ n× [1389.46]	4.54/ 4.57	48.37/ 48.41	2.74/ 2.76	8.05/ 8.07

Soluble (^a^) and insoluble (^b^) products of the reaction (1).

**Table 3 polymers-13-00829-t003:** Thermogravimetric analysis data of the II and its hexamer.

Compound	Temperature Range (°C)	DTG Peak (°C)	TG Weight Loss (%)		Assignment
			Calcul.	Experim.	
II	20–200	110	2.64	2.78	uncoordinated water (2 mole)
	200–500	320.9 425.2	7.20 23.51	8.32 23.07	dehydroxylation and deamination destruction of sulfo groups
	500–800	690.1 820.9	22.35 10.88	20.80 12.02	oxidation of the Ph-fragment of porphyrins oxidation of the Ph- fragment of ligands
	>900		33.41	33.01	(SnC20H12N4O2 rest)
[II-Cu]_n_	20–200	100 180	2.46 2.46	2.32 2.56	uncoordinated water (2 mole) coordinated water (2 mole)
	200–500	352.9 425.2	6.71 21.91	7.23 18.47	dehydroxylation and deamination destruction of sulfo groups
	500–800	694.5 870.2	20.83 10.14	22.30 8.69	oxidation of the Ph-fragment of porphyrins oxidation of the Ph- fragment of ligands
	>900		36.57	38.43	SnC_20_H_12_N_4_O_2_, CuO rest

**Table 4 polymers-13-00829-t004:** UV-Vis spectra of the studied compounds (I, II, Cu-[I-Cu]_6_, and Cu-[II-Cu]_6_).

Compounds	UV-Vis Spectra, λ_nm_(lgε)
I	419 (5.04), 555 (4.06), 594 (3.57)
I-Cu-I	418 (5.00), 554 (3.87), 595 (3.45), 610 (3.33)
Cu-[I-Cu]_6_	418 (4.98), 554 (3.78), 595 (3.40), 610 (3.89)
II	419 (5.11), 554 (4.10), 593 (3.61)
II-Cu-II	418 (5.05), 553 (4.07), 592 (3.48), 609 (3.29)
Cu-[II-Cu]_6_	418 (5.03), 553 (4.07), 592 (3.35), 609 (3.54)

**Table 5 polymers-13-00829-t005:** Relevant IR bands for the compounds I and Cu-[I-Cu]_6_.

I	Cu- [I-Cu]_6_	I	Cu- [I-Cu]_6_	II	Cu- [II-Cu]_6_	II	Cu- [II-Cu]_6_
NH^3+^ NH_2_		COO- COO-		N-H N-H		O-H O-H	
3188ν 3299ν 1655δ_d_ 1517δ_s_ 1246γ_r_ 1181γ_r_	3201ν 3230ν 1668δ_d_ 1534 δ 1200γ 1166γ	1607ν_as_ 1384ν_s_ 646δ_as_ 580δ_s_	1660ν_as_ 1405ν_s_ 606δ 588δ	3357ν 1619δ 764γ_w_	3430ν 1638δ 747γ	3244ν 1378δ_d_	- -
						C-O C-O	
						1152ν	1114ν
Cu-N Cu-N		Cu-O Cu-O		Cu-N Cu-N		Cu-O Cu-O	
-	633ν	-	472ν	-	620ν	-	480ν

**Table 6 polymers-13-00829-t006:** Relevant ^1^H-NMR signals for studied compounds.

Type of Protons	Chemical Shifts of Signals			Type of Protons	Chemical Shifts of Signals		
	I	I-Cu-I	Cu[I-Cu]_6_		II	II-Cu-II	Cu[II-Cu]_6_
-COOH	11.37 (s, 2H)	11.35 (s, H)	-	-OH	10.7 (s, 2H)	10.6 (s, H)	-
-NH_2_	6.72 (s, 4H)	6.71 (s, 2H), 6.91(brs, 2H)	6.93 (brs, 2H)	-NH_2_	8.59 (s, 4H)		8.79 (brs, 2H)
-CH(L)	4.37 (t, 2H)	4.36 (t, H) 3.81 (t, H)	3.78 (t, 2H)	-NH_2_	5.32 (s, 4H)	5.35 (s, 4H)	5.36 (s, 4H)
-CH_2_-	3.19 (m, 4H)	3.15 (m, 4H)	3.11 (m, 4H)	Ph(L)	5.97 (t, 2H)	5.99(m, 2H)	6.03 (t, 2H)
2-Ph (L)	5.51 (d, 4H)	5.64 (m, 4H)	5.82 (d, 4H)	Ph(L)	2.92 (t, 2H)	2.92 (t, 2H)	2.91 (t, 2H)
3-Ph (L)	2.28 (d, 4H)	2.30 (d, 4H)	2.35 (d, 4H)	2-Ph(Porph.)	8.45 (d, 8H)	8.46 (d, 8H)	8.44 (d, 8H)
2-Ph(Porph)	8.36 (d, 8H)	8.37 (d, 8H)	8.38 (d, 8H)	3-Ph(Porph.)	8.25 (d, 8H)	8.24(d, 8H)	8.23 (d, 8H)
3-Ph(Porph)	8.14 (d, 8H)	8.17 (d, 8H)	8.15 (d, 8H)	β-Por	9.10 (s, 8H)	9.13 (s, 8H)	9.12 (s, 8H)
β-Porph.	9.41(s, 8H)	9.43 (s, 8H)	9.42 (s, 8H)				

**Table 7 polymers-13-00829-t007:** Diffusion coefficients (D×10^−10^, m^2^s^−1^) of the complexes I and II and the products of their interaction with Cu^2+^ at the 1:1 and 1:5 concentration ratios of the reagents.

I	I-Cu-I	I-[Cu-I]_n_
2.96	2.26	1.55
**II**	**II-Cu-II**	**II-[Cu-II]_n_**
2.80	2.13	1.32

The measurement error is equal to ±0.04 ÷ 0.09 × 10^−10^, m^2^s^−1^.

## Data Availability

Samples of the compounds are available from the authors.

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
