# Peer review of "New Polyporphyrin Arrays with Controlled Fluorescence Obtained by Diaxial Sn(IV)-Porphyrin Phenolates Chelation with Cu2+ Cation"

_polymers, 2021, doi:10.3390/polym13050829_

Round 1
Reviewer 1 Report
This paper describes the syntheses, identifications, and fluorescent properties of oligomer or polymer of Sn(IV) porphyrins bridged by square planer Cu(II). The results were supported by theoretical calculations. The findings are interesting. However, this paper contains several careless mistakes. Thus, this paper is worth publishing in Polymers with minor revisions. Some additional comments are listed below.
1) The expression “axial complex” is hard to understand before looking at Figure 1. I recommend to use more clear expression. In addition, the expression of “axial coordination” in abstract and introduction is unsuitable.
2) There are many hyphenation mistakes as follows.
tetra-(4-sulfonatophenyl)-porphyrin --> tetra(4-sulfonatophenyl)porphyrin
red-ox --> redox
physico-chemical --> physicochemical
photo-physical --> photophysical
photo-activity --> photoactivity
hydroxy-group --> hydroxy group
anino-group --> amino group
high-purity --> high purity
quantum-chemical --> quantum chemical
mass-spectrometry --> mass spectrometry
sulfo-group --> sulfo group
square-planar --> square planer
3) page 3, line 98: What is porphyrin III?
4) page 8, line252-253: metal-ligand --> metal—ligand (hyphen --> em dash)
5) page 8, line 274: given in.47 --> given in [47].
6) page 10: Figure 5 is wrong. This figure should be replaced by the correct PXRD figure.
7) page 11, lines 322 and 326: amino- and carboxy-groups --> amino and carboxyl groups
8) page 12, line 357: planar-square --> square planer
9) page 14, line 396: Diffusion-ordered DOSY spectroscopy --> Diffusion ordered spectroscopy (DOSY)
10) page 14: Lines 408—419 should be removed because this part is the same with lines 396—407.
Author Response
Dear Colleague!
In accordance with your comments, we have made corrections in the text of the article (see attachment). With best respect, the authors.

Reviewer 2 Report
The manuscript by Mamardashvili and co-workers illustrate controlled fluorescence of novel polyporphyrin, Sn(IV)-porphyrin axial complexes with Cu2+ ions. The structure and properties of the synthesized porphyrin arrays are investigated quantum-chemical calculations and various analytical techniques. This work is of interest to readers and could be suitable for polymers after consideration of the following points:
1. For the synthesis of II, did the authors used basic, neutral or acidic alumina for chromatography.
2. The authors should check the figure 5 caption, PXRD plot should be intensity vs 2theta.
3. In figure 6, the authors provide plausible reasons of new shoulder at 615 nm, which is absent in I.
4. Along with DOSY, the Jobs plot should be used to highlight the concentration of two binding partners.
Author Response
Dear Colleague! In accordance with your comments, we have made corrections in the text of the article (see attachment).
With best respect, authors.

Reviewer 3 Report
The authors described a new coordination polymer/oligomer from Sn(IV)-tetra-(4-sulfonatophenyl)-porphyrin and Cu ions in this manuscript. The synthesis and structure characterization of the polymer/oligomer is well presented. The authors also showed the increased fluorescence after the chelation of porphyrinate axial complex with Cu ion. I recommend its publication in Polymers. However, a few comments are given below.
- Figure 5 is missing, please check the raw data.
- Please also indicate the sample name for both cuvettes in Figure 6.
- It’s not clear about the statement of “fast fluorescence increase” since there is no kinetics study in the manuscript.
- Please give the full name of the “TG” “DTA” “EPR”.
Author Response
Dear Colleague!
In accordance with your comments, we have made corrections in the text of the article (see attachment). With best regards, the authors.
